# Amyotrophic lateral sclerosis: Correlations between fluid biomarkers of NfL, TDP-43, and tau, and clinical characteristics

Yuta Kojima[1], Takashi Kasai[1]*, Yu-ichi Noto[1], Takuma Ohmichi[1], Harutsugu Tatebe[2], Takamasa Kitaoji[1], Yukiko Tsuji[1], Fukiko Kitani-Morii[1], Makiko Shinomoto[1], David Allsop[3†], Satoshi Teramukai[4], Toshiki Mizuno[1], Takahiko Tokuda[2]*

1 Department of Neurology, Kyoto Prefectural University of Medicine, Kyoto, Japan, 2 Department of Functional Brain Imaging Research, National Institute of Radiological Sciences, National Institutes for Quantum and Radiological Science and Technology, Chiba, Japan, 3 Faculty of Health and Medicine, Division of Biomedical and Life Sciences, Lancaster University, Lancaster, United Kingdom, 4 Department of Biostatistics, Kyoto Prefectural University of Medicine, Kyoto, Japan

† Deceased.
* kasaita@koto.kpu-m.ac.jp (TK); tokuda.takahiko@qst.go.jp (TT)

**Data Availability Statement:** All relevant data are within the paper and its Supporting Information files.

## Abstract

### Objectives

We previously reported the diagnostic and prognostic performance of neurofilament light chain (NfL), TAR DNA-binding protein 43 (TDP-43), and total tau (t-tau) in cerebrospinal fluid (CSF) and plasma as amyotrophic lateral sclerosis (ALS) biomarkers. The present study aimed to elucidate associations between clinical characteristics and the markers as well as mutual associations of the markers in ALS patients using the same dataset.

### Methods

NfL, TDP-43, and t-tau levels in CSF and plasma in 75 ALS patients were analyzed. The associations between those markers and clinical details were investigated by uni- and multi-variate analyses. Correlations between the markers were analyzed univariately.

### Results

In multivariate analysis of CSF proteins, the disease progression rate (DPR) was positively correlated with NfL (β: 0.51, p = 0.007) and t-tau (β: 0.37, p = 0.03). Plasma NfL was correlated with age (β: 0.53, p = 0.005) and diagnostic grade (β: -0.42, p = 0.02) in multivariate analysis. Plasma TDP-43 was correlated negatively with split hand index (β: -0.48, p = 0.04) and positively with % vital capacity (β: 0.64, p = 0.03) in multivariate analysis. Regarding mutual biomarker analysis, a negative correlation between CSF-NfL and TDP-43 was identified (r: -0.36, p = 0.002).

### Conclusions

Elevated NfL and t-tau levels in CSF may be biomarkers to predict rapid DPR from onset to sample collection. The negative relationship between CSF NfL and TDP-43 suggests that

**Funding:** This work was supported mainly by a grant from the Japan Agency for Medical Research and Development (AMED) (to T.T.) and in part by Grants-in-Aid (Nos. 15K09319 and 18K07506 to T. K., 20K16605 to T.O, and 18K15461 to H.T.) from the Ministry of Education, Culture, Sports, Science and Technology of Japan. The funders had no role in study design, data collection and analysis, decision to publish, or preparation of the manuscript.

**Competing interests:** The authors have declared that no competing interests exist.

elevation of CSF TDP-43 in ALS is not a simple consequence of its release into CSF during neurodegeneration. The negative correlation between plasma TDP-43 and split hand index may support the pathophysiological association between plasma TDP-43 and ALS.

## Introduction

Amyotrophic lateral sclerosis (ALS) is a neurodegenerative disorder specifically affecting motor neurons, characterized by progressive muscle weakness and atrophy, resulting in respiratory failure and death. It is untreatable at present, but an effective therapy is strongly desired [1]. In order to develop novel treatments, objective and responsive biomarkers are essential for successfully carrying out clinical trials. Various potential biomarkers of ALS have been reported, and recently neurofilament light chain (NfL), which is associated with neuroaxonal damage and loss [2], is considered the most promising biomarker of ALS progression [3–6]. The level of TAR DNA-binding protein 43 (TDP-43) was also reported to be high in the cerebrospinal fluid (CSF) and plasma of ALS patients compared with controls [7,8], but the inconsistency of the TDP-43 immunoassay used to measure the level of this protein has been problematic. CSF total-tau (t-tau) is expected to serve as a responsive biomarker for ALS diagnosis and progression, despite a lack of consistent results [9,10]. The diagnostic and prognostic value of such biomarkers has been well-investigated at individual levels; however, associations between biomarkers and clinical characteristics, as well as the mutual associations of those biomarkers, are not fully understood.

We previously examined the diagnostic and prognostic performance of NfL, TDP-43, and t-tau in CSF and plasma as biomarkers of ALS in two individual case-control cohorts, revealing increased levels of CSF NfL, plasma NfL, and CSF TDP-43 in ALS compared with control groups and shorter survival associated with higher levels of CSF and plasma NfL [11]. We herein conducted additional analysis of the same cohorts of ALS patients and biomarker dataset with the aim of comprehensively elucidating the association with clinical findings and mutual associations of CSF and plasma NfL, TDP-43, and t-tau levels in ALS patients, because such clinico-biomarker and mutual-biomarker associations were insufficiently investigated in the initial study.

## Materials and methods

### Patients and data collection

The study cohort consisted of 75 ALS patients visiting our institute between September 2009 and May 2018, who were from the discovery cohort enrolled from September 2009 to March 2014 (n = 29) and validation cohort enrolled from April 2014 to May 2018 (n = 46) included in our previous study [11]. The participating ALS patients were diagnosed with suspected, possible, probable, or definite ALS based on El Escorial criteria [12]. All ALS patients in the suspected or possible grade were converted to probable or definite ALS within the follow-up period. All participants underwent CSF sampling at the time of diagnosis. At the time of CSF collection, clinical data including age, sex, upper motor neuron (UMN) score, Medical Research Council (MRC) sum score, % slow vital capacity (%VC), ALS Functional Rating Scale revised (ALSFRS-R) [13], disease progression rate (DPR), and split hand index were collected. The UMN score comprised a sum of pathologically brisk reflexes that included bilateral assessment of the biceps, supinator, triceps, finger, knee, and ankle reflexes, plantar responses,

and facial and jaw jerks, with a maximum score of 16 [14]. The MRC sum score, which is a sensitive indicator of the degree of muscle weakness as a lower motor neuron symptom, was the total score of manual muscle testing on bilateral arm abduction, elbow flexion, wrist extension, hip flexion, knee extension, and ankle dorsiflexion, and the score ranged from 0 to 60 [15]. The DPR was calculated as follows: (48 − ALSFRS-R score at the time of sampling)/ months elapsed between disease onset and sampling [16]. Disease onset was defined as the month when patients become aware of their symptoms caused by ALS. Based on the nerve conduction study, split hand index was calculated by dividing the product of the compound muscle action potential (CMAP) amplitude recorded over the first dorsal interosseous and abductor pollicis brevis by the CMAP amplitude recorded over the abductor digiti minimi on the affected side [17] (Note: a reduced split hand index indicates preferential atrophy in the first dorsal interosseous and abductor pollicis brevis, which is typically observed in ALS).

All study subjects provided written informed consent before participation and the study protocol was approved by the University Ethics Committee (ERB-G-12). Informed consent from patients was obtained when possible and also from the nearest relative.

## Sample collection and measurement of NfL, TDP-43, and t-tau

CSF samples were collected in polypropylene vials by lumbar puncture, and the samples were cleared by centrifugation at 400 x g for 10 min at 4˚C immediately after collection. Plasma samples were obtained via venous puncture and collected in EDTA-containing tubes. After collection, plasma was separated by centrifugation for 10 min at 3,000 rpm and placed in polypropylene vials. CSF and plasma samples obtained from the enrolled subjects were immediately stored at -80˚C until analysis. In the analysis, CSF and plasma NfL, TDP-43, and t-tau concentrations were measured with Simoa NF-light assay, TDP-43 assay, and Human Total Tau assay kits, respectively, on a Simoa HD-1 Analyzer (Quanterix, Lexington, MA, USA) according to protocols provided by the manufacturer. All samples were analyzed in duplicate on one occasion independently in the two cohorts (discovery and validation cohorts). Because of inter-assay variation between the two cohorts, we corrected the values based on those of internal controls [11]. For this sub-analysis in the present study, we used the same levels of those combined values of each biomarker in the ALS group of our previous paper. In the original dataset, we conducted a case-control study involving two independent cohorts (validation and discovery cohorts). To correct for inter-assay variation, we adjusted the values of the validation cohort based on the correction formula: raw values x correction factors. In the current study, we used the corrected biomarker levels for the analysis of the two cohorts. The correction factors were determined as the mean value ratios between the discovery and validation assays based on four internal controls for each biomarker (See the previous study [11]).

## Statistics

Associations between biomarker values and clinical characteristics were analyzed by uni- and multivariate linear regression models using each biomarker value as a dependent variable, and age, sex, diagnosis, UMN score, MRC score, %VC, ALSFRS-R score, DPR, and split hand index as independent variables. The independent variables in multivariate analysis were chosen based on the importance in the association with the prognosis of ALS, not by p- values of univariate analysis or step-wise method, unless otherwise stated. Multivariate linear regression was performed after the exclusion of patients whose data were missing. The F test was used to assess how each multivariate linear regression model fitted the data. When dependent variables or residuals in multiple linear regression were not normally distributed (P<0.05 in the Shapiro-Wilk test), we reanalyzed those after natural logarithm transformation of dependent

variables. The absence of multicollinearities among these variables was confirmed when the value of the variance inflation factor was less than ten. The association between biomarkers was analyzed by Spearman's rank correlation tests. $P < 0.05$ was considered significant in all analyses. JMP software 14.0 (SAS Institute, Cary, North Carolina, USA) was used for all statistical analyses.

## Results

### Patient characteristics

The demographic characteristics of 75 patients with ALS are summarized in Table 1. For detailed clinical data from each patient, see S1 Table. Case numbers 1–29 were in the discovery cohort of our previous report and case numbers 30–75 were in the validation cohort [11]. This cohort included four familial ALS cases in which relatives with ALS or motor neuron disease were found within the second degree by consanguinity: cases 3 (younger brother), 8 (maternal grandmother), 41 (younger brother), and 57 (mother) (S1 Table). In those familial cases, no genetic abnormality was identified.

### Association between levels of biomarkers and clinical data

Univariate and multivariate analyses are shown in Tables 2–4. Levels of NfL, TDP-43, and t-tau in plasma were analyzed after natural logarithm transformation because they were not normally distributed.

As shown in Table 2A and 2B, NfL in CSF was significantly correlated with the diagnostic grade ($\beta$ = -0.34, p = 0.004), UMN score ($\beta$ = 0.34, p = 0.005), and DPR ($\beta$ = 0.44, p = 0.0001) in univariate analysis. A significant correlation in multivariate analysis was observed only between CSF NfL and DPR ($\beta$ = 0.51, p = 0.007). NfL in plasma showed significant correlations with age ($\beta$ = 0.45, p = 0.001), diagnostic grade ($\beta$ = -0.39, p = 0.006), %VC ($\beta$ = -0.42, p = 0.004), and DPR ($\beta$ = 0.29, p = 0.04) in univariate analysis. NfL in plasma was correlated

**Table 1. Clinical and biological data of patients with amyotrophic lateral sclerosis.**

| Characteristic | |
|---|---|
| Age at sample collection (years) | 72 [63–77] |
| Sex (Male:Female) (n) | 47:28 |
| *Clinical diagnosis at sample collection (Definite:Probable:Possible:Suspected) (n) | 25:28:15:7 |
| ALSFRS-R score (n = 75) | 41 [35 – 44] |
| Disease progression rate (n = 75) | 0.47 [0.16–1.00] |
| Upper motor neuron score (n = 75) | 8 [4 – 11] |
| MRC sum score (n = 75) | 52 [47–57] |
| % vital capacity (%) (n = 69) | 87 [73–103] |
| Split hand index (n = 62) | 5.0 [2.0–10.0] |
| Neurofilament light chain in cerebrospinal fluid (pg/mL) (n = 70) | 7965 [4261–13433] |
| TAR DNA-binding protein 43 in cerebrospinal fluid (pg/mL) (n = 70) | 63 [60–68] |
| Total-tau in cerebrospinal fluid (pg/mL) (n = 69) | 12 [11 – 16] |
| Neurofilament light chain in plasma (pg/mL) (n = 49) | 109 [68–160] |
| TAR DNA-binding protein 43 in plasma (pg/mL) (n = 49) | 391 [215–761] |
| Total-tau in plasma (pg/mL) (n = 48) | 0.63 [0.40–0.86] |

Data are given as median [interquartile range].

ALSFRS-R, Amyotrophic Lateral Sclerosis Functional Rating Scale revised score; MRC, Medical Research Council.

* Patients were diagnosed based on El Escorial criteria.

**Table 2. A: The association between NfL in CSF and clinical characteristics of ALS patients (n = 70).** B: The association between NfL in plasma and clinical characteristics of ALS patients (n = 49).

| | Univariate analysis | | | | Multivariate analysis (N = 53, $R^2$ = 0.40, P = 0.005) | | | |
|---|---|---|---|---|---|---|---|---|
| | **B value** | **SE** | **β value** | **P-value** | **B value** | **SE** | **β value** | **P-value** |
| Age | 27 | 69 | 0.05 | 0.70 | -87 | 75 | -0.17 | 0.25 |
| Sex* | -1468 | 1568 | -0.11 | 0.35 | -368 | 1689 | -0.03 | 0.83 |
| Diagnosis** | **-4682** | **1554** | **-0.34** | **0.004** | -3123 | 1864 | -0.25 | 0.10 |
| Upper motor neuron score | **454** | **154** | **0.34** | **0.005** | 172 | 181 | 0.14 | 0.35 |
| MRC score | -87 | 92 | -0.11 | 0.35 | -247 | 149 | -0.26 | 0.11 |
| % vital capacity | -21 | 35 | -0.07 | 0.56 | 67 | 49 | 0.22 | 0.18 |
| ALSFRS-R score | -182 | 111 | -0.2 | 0.11 | 261.00 | 241 | 0.25 | 0.28 |
| Disease progression rate | **4458** | **1097** | **0.44** | **0.0001** | **5418** | **1908** | **0.51** | **0.007** |
| Split hand index | -303 | 163 | -0.24 | 0.07 | -75.00 | 154 | -0.06 | 0.63 |

* Male = 0; Female = 1.

**Definite/Probable = 0; Possible/Suspected = 1.

NfL, neurofilament light chain; CSF, cerebrospinal fluid; ALS, amyotrophic lateral sclerosis.

MRC, Medical Research Council; ALSFRS-R score, ALS Functional Rating Scale revised score.

Bold characters indicate a significant difference with a p-value less than 0.05.

The data were analyzed after being log transformed.

| | Univariate analysis | | | | Multivariate analysis (n = 32, $R^2$ = 0.61, P = 0.006) | | | |
|---|---|---|---|---|---|---|---|---|
| | B value | SE | β value | P-value | B value | SE | β value | P-value |
| Age | **0.03** | **0.009** | **0.46** | **0.0009** | **0.03** | **0.009** | **0.53** | **0.005** |
| Sex* | -0.39 | 0.26 | -0.21 | 0.15 | -0.14 | 0.22 | -0.10 | 0.52 |
| Diagnosis** | **-0.98** | **0.26** | **-0.48** | **0.0005** | **-0.68** | **0.27** | **-0.42** | **0.02** |
| Upper motor neuron score | 0.06 | 0.03 | 0.27 | 0.06 | 0.04 | 0.03 | 0.25 | 0.11 |
| MRC score | -0.01 | 0.01 | -0.09 | 0.55 | -0.01 | 0.02 | -0.11 | 0.56 |
| % vital capacity | **-0.01** | **0.005** | **-0.44** | **0.003** | -0.004 | 0.007 | -0.12 | 0.61 |
| ALSFRS-R score | -0.02 | 0.02 | -0.22 | 0.13 | 0.05 | 0.04 | 0.39 | 0.19 |
| Disease progression rate | 0.23 | 0.13 | 0.25 | 0.08 | 0.36 | 0.24 | 0.30 | 0.16 |
| Split hand index | **-0.06** | **0.02** | **-0.37** | **0.03** | -0.02 | 0.02 | -0.12 | 0.49 |

* Male = 0; Female = 1.

**Definite/Probable = 0; Possible/Suspected = 1.

NfL, neurofilament light chain; ALS, amyotrophic lateral sclerosis; MRC, Medical Research Council; ALSFRS-R score, ALS Functional Rating Scale revised score.

Bold characters indicate a significant difference with a p-value less than 0.05.

with age (β = 0.53, p = 0.005) and diagnostic grade (β = -0.42, p = 0.02) in multivariate analysis. Regarding the goodness of fit in the multivariate linear regression model, the model significantly fitted the data on NfL in both CSF and plasma.

Regarding CSF TDP-43, there was no significant correlation on univariate analysis. The level in CSF was significantly negatively correlated with DPR in multivariate analysis (β = -0.47, p = 0.03). (Note, this weak correlation cannot be generalized because of the small $R^2$ value and lack of significance of model-fitting in the F test.) (Table 3A) TDP-43 in plasma was significantly positively correlated with % vital capacity (β = 0.34, p = 0.03) and ALSFRS-R score (β = 0.33, p = 0.02) in univariate analysis. The significant correlation between TDP-43 in plasma and %VC was preserved in multivariate analysis (β = 0.64, p = 0.03). Multivariate analysis revealed another significant negative correlation between TDP-43 in plasma and split hand index (β = -0.61, p = 0.008), as shown in Table 3B. Although the goodness of fit in the

**Table 3. A: The association between TDP-43 in CSF and clinical characteristics of ALS patients (n = 70).** B: The association between TDP-43 in plasma and clinical characteristics of ALS patients (n = 49).

| | Univariate analysis | | | | Multivariate analysis (n = 53, $R^2$ = 0.15, P = 0.59) | | | |
|---|---|---|---|---|---|---|---|---|
| | **B value** | **SE** | **β value** | **P-value** | **B value** | **SE** | **β value** | **P-value** |
| Age | 0.05 | 0.13 | 0.05 | 0.69 | -0.01 | 0.18 | -0.01 | 0.98 |
| Sex* | 3.46 | 2.84 | 0.15 | 0.23 | 7.09 | 4.05 | 0.29 | 0.09 |
| Diagnosis** | 0.46 | 3.01 | 0.02 | 0.88 | -0.09 | 4.48 | -0.003 | 0.98 |
| Upper motor neuron score | 0.01 | 0.30 | 0.004 | 0.97 | 0.02 | 0.44 | 0.01 | 0.96 |
| MRC score | 0.01 | 0.17 | 0.009 | 0.94 | 0.35 | 0.36 | 0.18 | 0.34 |
| % vital capacity | -0.02 | 0.06 | -0.04 | 0.74 | 0.06 | 0.12 | 0.09 | 0.64 |
| ALSFRS-R score | -0.11 | 0.20 | -0.07 | 0.57 | -0.96 | 0.58 | -0.46 | 0.10 |
| Disease progression rate | -2.00 | 2.21 | -0.11 | 0.37 | **-10.2** | **4.58** | **-0.47** | **0.03** |
| Split hand index | 0.05 | 0.33 | 0.02 | 0.89 | -0.01 | 0.37 | -0.004 | 0.98 |

* Male = 0; Female = 1.

**Definite/Probable = 0; Possible/Suspected = 1.

TDP-43, TAR DNA-binding protein 43; CSF, cerebrospinal fluid; ALS, amyotrophic lateral sclerosis; MRC, Medical Research Council; ALSFRS-R score, ALS Functional Rating Scale revised score.

Bold characters indicate a significant difference with a p-value less than 0.05.

| | Univariate analysis | | | | Multivariate analysis (n = 32, $R^2$ = 0.42, P = 0.13) | | | |
|---|---|---|---|---|---|---|---|---|
| | B value | SE | β value | P-value | B value | SE | β value | P-value |
| Age | -0.01 | 0.01 | -0.14 | 0.33 | -0.01 | 0.02 | -0.17 | 0.42 |
| Sex* | 0.12 | 0.32 | 0.06 | 0.70 | 0.16 | 0.36 | 0.08 | 0.66 |
| Diagnosis** | 0.52 | 0.34 | 0.22 | 0.14 | 0.66 | 0.45 | 0.29 | 0.16 |
| Upper motor neuron score | 0.04 | 0.03 | 0.17 | 0.25 | 0.03 | 0.04 | 0.11 | 0.54 |
| MRC score | 0.03 | 0.02 | 0.26 | 0.07 | 0.04 | 0.03 | 0.24 | 0.29 |
| % vital capacity | **0.01** | **0.006** | **0.34** | **0.03** | **0.03** | **0.01** | **0.64** | **0.03** |
| ALSFRS-R score | **0.04** | **0.02** | **0.33** | **0.02** | -0.10 | 0.06 | -0.60 | 0.10 |
| Disease progression rate | -0.10 | 0.15 | -0.09 | 0.53 | -0.36 | 0.41 | -0.22 | 0.39 |
| Split hand index | -0.02 | 0.03 | -0.13 | 0.46 | **-0.12** | **0.04** | **-0.61** | **0.008** |

The data were analyzed after being log transformed.

* Male = 0; Female = 1.

**Definite/Probable = 0; Possible/Suspected = 1.

TDP-43, TAR DNA-binding protein 43; ALS, amyotrophic lateral sclerosis; MRC, Medical Research Council; ALSFRS-R score, ALS Functional Rating Scale revised score.

Bold characters indicate a significant difference with a p-value less than 0.05.

multivariate regression model for plasma TDP-43 did not reach significance in the F test, the $R^2$ value of the model (= 0.42) was not small. Moreover, when we employed a backward step-wise method for modeling, a significant model using age, diagnosis, MRC score, ALSFRS-R, % vital capacity, and split hand index, was obtained ($R^2$ value = 0.36; p-value in F test = 0.042). The significant correlation of levels of plasma TDP-43 with %VC and split hand index was preserved in this model (β = -0.64, p = 0.021 and β = -0.56, p = 0.006, respectively). Based on those results from forced entry and backward step-wise methods in multiple regression, we concluded that the correlations between plasma TDP-43 and %VC as well as between plasma TDP-43 and split hand index show statistical reliability.

On analyzing t-tau, there were significant correlations of t-tau in CSF with age (β = 0.32, p = 0.007) and sex (β = 0.44, p = 0.0001) in univariate analysis, and with sex (β = 0.46, p = 0.0009) and DPR (β = 0.37, p = 0.03) in multivariate analysis with a significant goodness of

**Table 4. A: The association between t-tau in CSF and clinical characteristics of ALS patients (n = 69).** B: The association between t-tau in plasma and clinical characteristics of ALS patients (n = 48).

| | Univariate analysis | | | | Multivariate analysis (n = 53, $R^2$ = 0.47, P = 0.0006) | | | |
| --- | --- | --- | --- | --- | --- | --- | --- | --- |
| | B value | SE | β value | P-value | B value | SE | β value | P-value |
| Age | **0.14** | **0.05** | **0.32** | **0.007** | 0.06 | 0.06 | 0.14 | 0.29 |
| Sex* | **4.43** | **1.09** | **0.44** | **0.0001** | **4.60** | **1.29** | **0.46** | **0.0009** |
| Diagnosis** | 0.07 | 1.28 | 0.007 | 0.95 | 0.29 | 1.43 | 0.03 | 0.84 |
| Upper motor neuron score | -0.04 | 0.13 | -0.04 | 0.77 | -0.13 | 0.11 | -0.12 | 0.38 |
| MRC score | -0.01 | 0.07 | -0.02 | 0.84 | 0.02 | 0.11 | 0.02 | 0.89 |
| % vital capacity | 0.01 | 0.03 | 0.03 | 0.83 | 0.06 | 0.04 | 0.25 | 0.10 |
| ALSFRS-R score | -0.01 | 0.09 | -0.01 | 0.93 | -0.02 | 0.18 | -0.03 | 0.91 |
| Disease progression rate | 1.70 | 0.92 | 0.22 | 0.07 | **3.23** | **1.46** | **0.37** | **0.03** |
| Split hand index | 0.08 | 0.13 | 0.09 | 0.52 | 0.13 | 0.12 | 0.13 | 0.29 |

* Male = 0; Female = 1.

**Definite/Probable = 0; Possible/Suspected = 1.

t-tau, total Tau; CSF, cerebrospinal fluid; ALS, amyotrophic lateral sclerosis; MRC, Medical Research Council; ALSFRS-R score, ALS Functional Rating Scale revised score.

Bold characters indicate a significant difference with a p-value less than 0.05.

| | Univariate analysis | | | | Multivariate analysis (n = 31, $R^2$ = 0.30, P = 0.48) | | | |
| --- | --- | --- | --- | --- | --- | --- | --- | --- |
| | B value | SE | β value | P-value | B value | SE | β value | P-value |
| Age | 0.009 | 0.008 | 0.17 | 0.25 | 0.01 | 0.01 | 0.29 | 0.22 |
| Sex* | -0.06 | 0.21 | -0.04 | 0.77 | -0.10 | 0.27 | -0.07 | 0.71 |
| Diagnosis** | 0.16 | 0.23 | 0.10 | 0.48 | -0.009 | 0.34 | -0.006 | 0.98 |
| Upper motor neuron score | 0.04 | 0.02 | 0.24 | 0.10 | 0.05 | 0.03 | 0.29 | 0.17 |
| MRC score | 0.009 | 0.01 | 0.12 | 0.40 | 0.03 | 0.02 | 0.31 | 0.23 |
| % vital capacity | -0.004 | 0.004 | -0.12 | 0.42 | -0.007 | 0.009 | -0.22 | 0.48 |
| ALSFRS-R score | -0.001 | 0.01 | -0.01 | 0.95 | 0.02 | 0.05 | 0.15 | 0.71 |
| Disease progression rate | 0.02 | 0.10 | 0.03 | 0.85 | 0.39 | 0.31 | 0.36 | 0.22 |
| Split hand index | 0.04 | 0.02 | 0.31 | 0.07 | 0.05 | 0.03 | 0.41 | 0.09 |

The data were analyzed after being log transformed.

* Male = 0; Female = 1.

**Definite/Probable = 0; Possible/Suspected = 1.

t-tau, total Tau; ALS, amyotrophic lateral sclerosis; MRC, Medical Research Council; ALSFRS-R score, ALS Functional Rating Scale revised score.

Bold characters indicate a significant difference with a p-value less than 0.05.

fit of the data (Table 4A). No association of t-tau in plasma with any clinical data was identified in either uni- or multivariate analysis (Table 4B).

## Correlation between levels of each biomarker

As shown in Fig 1A, NfL levels in CSF were negatively correlated with TDP-43 levels in CSF (r = -0.36, p = 0.002). There were no correlations between t-tau and NfL levels or between t-tau and TDP-43 levels in CSF (Fig 1B (t-tau—NfL); r = 0.14, p = 0.26. Fig 1C (t-tau—TDP-43); r = 0.02, p = 0.87). Mutual associations of the plasma biomarkers are summarized in Fig 1D–1F. There were no correlations in any combinations of the plasma biomarkers (NfL—TDP-43; r = -0.14, p = 0.35. t-tau—NfL; r = 0.09, p = 0.53. t-tau—TDP-43; r = -0.04, p = 0.80) (Note: correlations between CSF and plasma levels of each biomarker were already described in our previous report [11]).

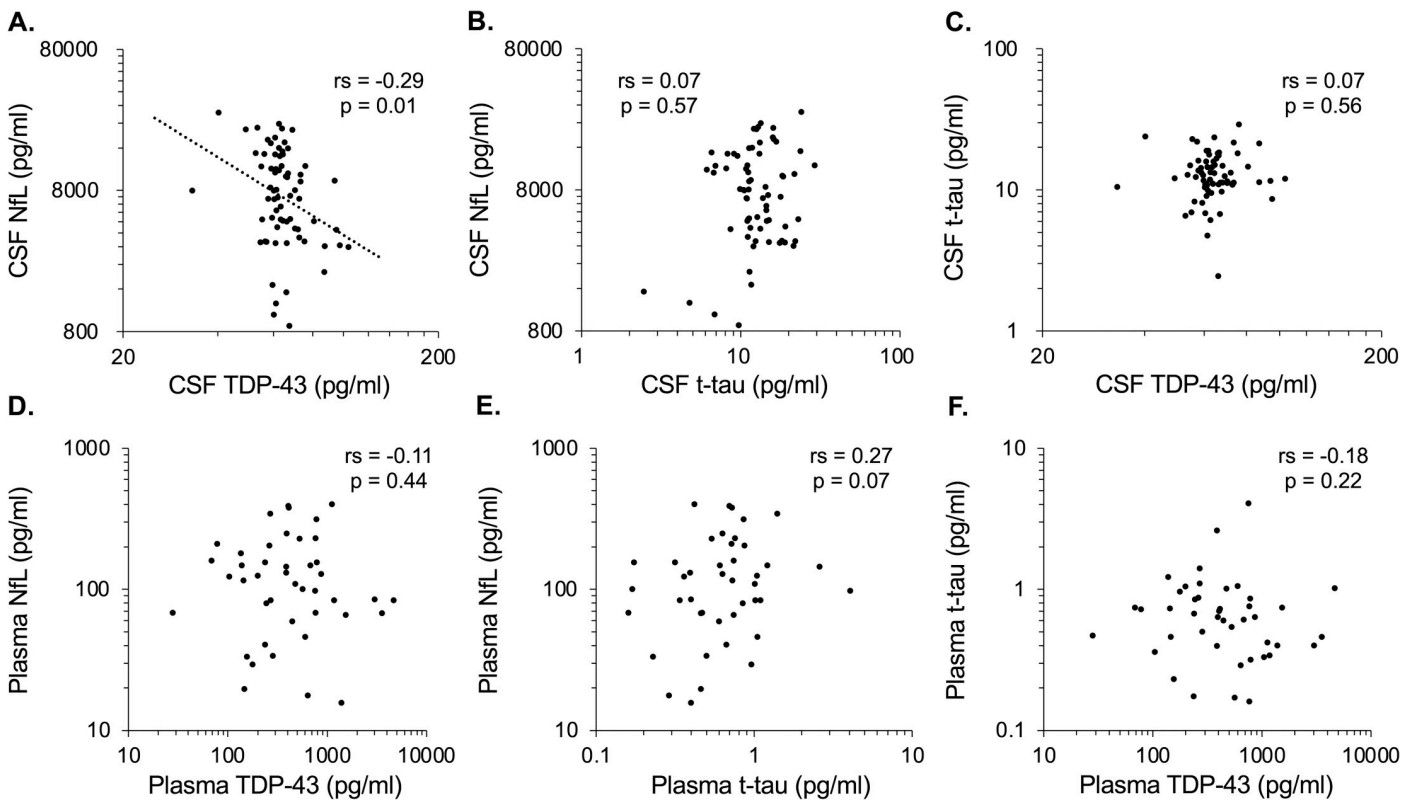

**Fig 1. Correlation of inter-biomarker levels in CSF and plasma.** Levels of individual biomarkers in ALS patients are presented as logarithms. The rs and p-values generated by Spearman's rank correlation tests are shown. The combinations of biomarkers in CSF are presented in A-C and those in plasma are in D-F. Correlations between the same markers in CSF and plasma were presented in our previous report [11]. A linear regression line is presented when there is a significant correlation. CSF, cerebrospinal fluid; NfL, neurofilament light; TDP-43, TAR DNA-binding protein 43; t-tau, total tau; ALS, amyotrophic lateral sclerosis.

## Discussion

NfL, one of the neuron-specific cytoskeletal proteins, is highly expressed in large-caliber myelinated axons. NfL is released into extracellular spaces and levels increase in blood and CSF proportionally to the degree of axonal damage [18]. Mutations in the NfL gene (*NEFL*: OMIM162280) cause hereditary neuropathies, while a direct association between ALS and *NEFL* mutation has not been reported [19]. TDP-43 is a heterogeneous nuclear ribonucleoprotein mainly distributed in the nucleus under physiological conditions. In most cases of ALS, it is translocated to the cytoplasm where it changes structure and forms aggregates. A direct mechanistic link between TDP-43 and neurodegeneration of motor neurons is suggested by the identification of mutations in the gene coding TDP-43 (*TARDBP*: OMIM605078) in hereditary forms of ALS [20]. Tau protein is mainly expressed in neurons of the central nervous system where it exerts a role in stabilizing microtubules, key components of axonal transport and signal transduction. Mutations of the Tau gene (MAPT: OMIM157140) are responsible for a rare form of familial dementia called frontotemporal lobar degeneration-17 with tau pathology, while such mutations of tau have not been found in familial ALS. The increase of CSF t-tau is probably due to its leakage from damaged neurons into CSF; therefore, it reflects the severity of neuronal damage and degeneration [21].

In the present study, we carried out a comprehensive investigation of associations of multiple biomarkers in CSF and plasma with clinical findings and the mutual associations of those biomarkers.

Regarding the associations of NfL and TDP-43 in CSF with clinical findings, the following two results were obtained: First, NfL in CSF was positively associated solely with DPR in multivariate analysis. The results were consistent with previous findings [22–26]. Second, in contrast, TDP-43 in CSF was negatively associated with DPR in multivariate analysis, although the fitness of the multiple regression was insufficient. This contrasting association of NfL and TDP-43 in CSF with DPR of ALS from onset to sample collection may explain the negative correlation between NfL and TDP-43 in CSF, which was only significant in mutual correlations of biomarkers in the present study. It is logical that NfL, known as a strong indicator of axonal damage, was correlated with the rapid progression of ALS, while the result that TDP-43 was associated with slow progression of the disease was unexpected. This fact suggests that CSF TDP-43 elevation in ALS is not a simple consequence of its release into CSF during neurodegeneration. Although the significance of increased CSF TDP-43 in ALS is still unclear, this phenomenon might be a result of increased levels of TDP-43 expression in motor neurons in the disease. Intranuclear TDP-43 negatively regulates its own expression; therefore, intranuclear loss of function of the molecule resulting from its abnormal aggregation and redistribution can lead to an elevation of its own expression and cytosolic protein levels [27,28]. In fact, increased levels of TDP-43 mRNA were observed in the brains of patients affected by various forms of FTLD with TDP-43 pathology [27,29]. Such overexpression of TDP-43 has been reported to lead to more serious protein-aggregation and loss of intranuclear function of the molecule, causing a vicious cycle [28]. However, only in the early stage of the disease, such overexpression and overproduction of TDP-43 in ALS might work as a compensatory mechanism to maintain its disturbed intranuclear function. This hypothetical idea might not only explain the current result of the association between elevated CSF TDP-43 and slow DPR in the early stage (between onset to sampling), but also be consistent with our previous finding that CSF TDP-43 was not associated with survival after sampling [11].

The TDP-43 level in plasma showed a negative association with split hand index and a positive association with %VC in the current study. The direct relevance of the association between the pathophysiology of ALS and changes in plasma TDP-43 levels still remains controversial. Actually, TDP-43 levels in plasma were elevated not only in patients with ALS and inclusion body myositis but also in those with other inflammatory myopathies, suggesting that increased levels of plasma TDP-43 may be due to the release of non-pathological TDP-43 as a result of muscle damage or inflammatory processes to some extent [30]. On the other hand, elevated plasma TDP-43 levels have been reproducibly observed in patients with ALS and FTLD [8,11,31], leading us to consider its disease-specificity for ALS. Our result that the decrease in split hand index, which is recognized as a specific and early-diagnostic marker of ALS [32], was related to TDP-43 elevation in plasma independently of other non-specific indicators of muscle weakness such as MRC score supports the suggestion that raised TDP-43 levels in plasma reflect the ALS pathophysiology.

T-tau in CSF showed positive associations with DPR of ALS and female sex. This is consistent with the previous observation that t-tau levels in CSF are elevated in many patients with chronic neurodegenerative diseases as a non-specific indicator of neuronal damage and axonal degeneration [33]. The result was also in agreement with our previous report on the same dataset, showing that the correlation between CSF t-tau and poor survival after sample collection was significant only in one of the two cohorts despite a lack of consistency between the cohorts [11]. Considering the conflicting results on the association between CSF t-tau levels and disease progression in ALS [9,10,34], the prognostic potential of this biomarker may not be as powerful as that of CSF NfL. Regarding the association with sex, Rosen et al. reported higher t-tau levels in CSF from female patients with Alzheimer's disease compared with male patients independent of the degree of cognitive impairment [35]. The sex difference in CSF t-tau has

been insufficiently considered in the biomarker field of ALS [9,36]. The fact that t-tau in CSF increased in females should be paid attention to if t-tau is used as a prognostic biomarker of ALS.

In univariate analysis, NfL in plasma showed significant correlations with age, diagnostic grade, %VC, and DPR. Age and DPR were also associated with plasma NfL in multivariate analysis (Table 2B). Of these, the associations with DPR and diagnostic grade were consistent with previous reports [22,24,37], while the result of age was different from other reports, which identified no relationships among these parameters [24,38]. This discrepancies in the results may be explained by the confounding factor of physiological age-dependent elevation of NfL [39].

Major limitations were that this was a single-center study with a relatively small sample size, which may have resulted in a weak statistical power. A point to be aware of regarding the multiple regression analyses was the predictive power of the independent variables for each biomarker level. "DPR for CSF NfL", "age and diagnostic grade for plasma NfL", and "sex and DPR for CSF t-tau" had high beta values in the multiple linear regression models with significant goodness of fit of the data. Meanwhile, the linear regression model for CSF TDP-43 had a relatively low $R^2$ value and did not significantly fit the data, although CSF TDP-43 was significantly correlated with a good prognosis. Given this fact, the predictive power of CSF TDP-43 for the progression of ALS was considered to remain uncertain. In addition, the longitudinal change of each biomarker could not be followed because CSF sampling was invasive and performed only once. The follow-up of biomarkers should be performed in a longitudinal study in order to clarify the significance of each biomarker in ALS progression.

## Conclusions

NfL levels in CSF were positively associated with DPR during the time from onset to sample collection and negatively associated with CSF TDP-43. This negative relationship suggests that elevation of CSF TDP-43 in ALS is not a simple consequence of its release into CSF during neurodegeneration. T-tau in CSF showed a positive association with DPR and female sex independently. These findings may be important for utilizing t-tau in CSF as a progression biomarker of ALS. A negative association between TDP-43 in plasma and split hand index was identified. However, the present results need to be validated in a larger cohort.

### Ethical publication statement

The authors confirm that they have read the journal's position on issues involved in ethical publication and affirm that this report is consistent with those guidelines. The patients were evaluated at Kyoto Prefectural University of Medicine hospital under a protocol approved by the local ethics committee of the university. Written informed consent was obtained from the patients.

## Supporting information

**S1 Table. Clinical information and concentrations of biomarkers.**
(DOCX)

## Author Contributions

**Conceptualization:** Yuta Kojima, Takashi Kasai, David Allsop, Takahiko Tokuda.

**Data curation:** Yuta Kojima, Takashi Kasai, Yu-ichi Noto, Takuma Ohmichi, Harutsugu Tatebe, Takamasa Kitaoji, Yukiko Tsuji, Fukiko Kitani-Morii, Makiko Shinomoto, Satoshi Teramukai, Takahiko Tokuda.

**Formal analysis:** Yuta Kojima, Takashi Kasai, Satoshi Teramukai, Takahiko Tokuda.

**Funding acquisition:** Takashi Kasai, Takahiko Tokuda.

**Investigation:** Takashi Kasai, Harutsugu Tatebe, Fukiko Kitani-Morii, Makiko Shinomoto, Takahiko Tokuda.

**Supervision:** David Allsop, Satoshi Teramukai, Toshiki Mizuno, Takahiko Tokuda.

**Writing – original draft:** Yuta Kojima.

**Writing – review & editing:** Yuta Kojima, Takashi Kasai, Yu-ichi Noto, Yukiko Tsuji, Fukiko Kitani-Morii, David Allsop, Satoshi Teramukai, Toshiki Mizuno, Takahiko Tokuda.

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
