## [Decision Letter · Decision Letter 0]

27 Aug 2021

PONE-D-21-22395

Amyotrophic lateral sclerosis: Correlations between fluid biomarkers of NfL, TDP-43, tau, and clinical characteristics

PLOS ONE

Dear Dr. Kasai,

Thank you for submitting your manuscript to PLOS ONE. After careful consideration, we feel that it has merit but does not fully meet PLOS ONE’s publication criteria as it currently stands. Therefore, we invite you to submit a revised version of the manuscript that addresses the points raised during the review process.

We look forward to receiving your revised manuscript.

Kind regards,

Weidong Le

Academic Editor

PLOS ONE

Journal Requirements:

“NO - Include this sentence at the end of your statement: The funders had no role in study design, data collection and analysis, decision to publish, or preparation of the manuscript.”

“This work was supported mainly by a grant from the Japan Agency for Medical Research and Development (AMED) (to T.T.) and in part by Grants-in-Aid (Nos. 15K09319 and 18K07506 to T.K., 20K16605 to T.O, and 18K15461 to H.T.) from the Ministry of Education, Culture, Sports, Science and Technology of Japan. The funders had no role in study design, data collection and analysis, decision to publish, or preparation of the manuscript”

“NO - Include this sentence at the end of your statement: The funders had no role in study design, data collection and analysis, decision to publish, or preparation of the manuscript.”

Reviewers' comments:

Reviewer's Responses to Questions

**Comments to the Author**

1. Is the manuscript technically sound, and do the data support the conclusions?

Reviewer #1: Partly

Reviewer #2: Yes

Reviewer #3: Yes

2. Has the statistical analysis been performed appropriately and rigorously? 

Reviewer #1: Yes

Reviewer #2: Yes

Reviewer #3: Yes

3. Have the authors made all data underlying the findings in their manuscript fully available?

Reviewer #1: Yes

Reviewer #2: Yes

Reviewer #3: Yes

4. Is the manuscript presented in an intelligible fashion and written in standard English?

Reviewer #1: Yes

Reviewer #2: Yes

Reviewer #3: Yes

5. Review Comments to the Author

Reviewer #1: This research is interesting, but there might be some errors and main concerns:

In P3, There might be an error. “Plasma TDP-43 correlated negatively with split hand index......” was shown in “Results”, but in “Conclusions”, it was shown “The positive correlation between plasma TDP-43 and split hand index......”.

This study cohort consisted of 75 ALS patients, it might be better for the authors to indicate the patients are only sporadic or sporadic and familial cases.

In P6, The authors indicated that “The participating ALS patients were diagnosed as suspected, possible, probable, or definite ALS based on revised El Escorial criteria.” . But in revised El Escorial criteria, there are four diagnostic categories, clinically definite, clinically probable, clinically probable–laboratory-supported and clinically possible. The category of “clinically suspected ALS was deleted from the revised El Escorial Criteria for the Diagnosis of ALS.

The authors used “% vital capacity” as a clinical data, did it mean “% forced vital capacity (FVC)”? And, if ALS patients had bulbar dysfunction, decreased % FVC could not reflect the real situation of respiratory function. Even if all participants had normal bulbar function, it might be better to analyze the relation of abnormal % FVC with those fluid biomarkers.

“progression rate” was used in this study, it might be better to use “disease progression rate (DPR)”.

“split hand index” was used as a clinical data in this study, but the hand muscles innervated by median and ulnar nerves might not be involved in some patients in this cohort. And it was shown “the split hand index was calculated by dividing the product of he compound muscle action potential (CMAP) amplitude recorded over the first orsal interosseous and abductor pollicis brevis by the CMAP amplitude recorded over the abductor digiti minimi ” in page 7, but the authors did not indicate that nerve conduction study was made in which side, left or right, or the involved side. These will certainly influence the research results.

In discussion, it might be better to discuss the probable reasons of the most important research results in more detail.

Reviewer #2: This is an elegant study on a fundamental topic in ALS biomarker research. The authors are experts in the field. The results are interesting and well presented. I have only some minor issues:

1. Line 91: the authors state that the patients of the discovery and replication cohorts were "extracted" from the cohorts of a previous study. It is not clear whether only some of the patients evaluated in the previous study were taken, and, if so, which criteria were applied for this selection.

2. Lines 134-136 ("All samples were analyzed in duplicate on one occasion. For this sub-analysis, we used the levels of each biomarker in the ALS group of our previous paper"). I think that these sentences are not so clear. Does this mean that for these patients the levels of the biomarkers were not measured again after the work made for the previous publication? Please clarify.

3. Line 190: "genialized"  "generalized"?

4. Lines 191-194 ("we would like to emphasize that CSF TDP-43 were negatively associated with the progression rate both in univariate and multivariate analyses regardless of significance, in contrast to those in CSF NfL and CSF tau"). I do not think that is correct to equalize a significant result (multivariate analysis) and a non-significant one. Perhaps the above-mentioned sentence could be eliminated.

5. Lines 353-354. Here it is written that the association between plasma TDP-43 and the split hand index is positive. However, I think that it is negative.

6. In table 1 I see that median t-Tau in CSF in ALS patients was 12 pg/mL. It seems a low value. Do the authors confirm this value?

Reviewer #3: The authors extended their analysis of CSF NfL and TDP-43 from a cohort of patients with ALS (published previously), by comparing measures to commonly used clinical measures of disease progression. Their methodology is clearly described and I have no concerns with their approach. I have no comments on their approach. This is sound. I have a few minor suggestions and some points of interest.

1. I initially suspected that the correlation between CSF NfL and TDP43 was driven by a few outliers. However after closer consideration (thank you for providing data points for individual patients) it appears that 5 of the participants present with very low CSF NfL levels relative to their TDP43 scores (patients 4, 9, 10, 17 and 23). Most of these individuals have very slow progression rates, except for #4, which has a very fast progression rate. Did the authors note any specific clinical features in these patients that may explain these low NfL scores. Could other clinical features be relevant? I note that all of these patients had a possible/probable diagnosis at study inclusion.

2. Did familial/sporadic status impact the results? Is the genetic status of patients known? If so, can the authors please comment.

3. I caution the authors against the use of “biomarker” for their current data results; factors are tested as possible prognostic indicators. This limitation is noted in the discussion, however the authors are asked to revise the term “biomarker” in their abstract.

4. Figure legend 1: Please correct the figure legend to state that correlations between plasma and CSF markers WERE completed in the previous manuscript. This should be followed by a reference to the report.

5. The manuscript is well-written and easy to follow, however I’ve noted a few minor grammatical mistakes; i.e. line 308 “missing the word “is” between CSF and elevated. The authors are encouraged to revise the document.

6. PLOS authors have the option to publish the peer review history of their article (what does this mean?). If published, this will include your full peer review and any attached files.

Reviewer #1: No

Reviewer #2: No

Reviewer #3: **Yes: **Frederik J Steyn

---

## [Author Response · Author response to Decision Letter 0]

5 Oct 2021

Responses to the reviewers’ comments

We appreciate the thoughtful comments from the reviewers that helped us to improve this manuscript. We have revised the manuscript with the point-by-point response as suggested, and believe that it has been significantly improved.

---

## [Decision Letter · Decision Letter 1]

8 Nov 2021

Amyotrophic lateral sclerosis: Correlations between fluid biomarkers of NfL, TDP-43, and tau, and clinical characteristics

PONE-D-21-22395R1

Dear Dr. Kasai,

We’re pleased to inform you that your manuscript has been judged scientifically suitable for publication and will be formally accepted for publication once it meets all outstanding technical requirements.

Kind regards,

Weidong Le

Academic Editor

PLOS ONE

Additional Editor Comments (optional):

Reviewers' comments:

Reviewer's Responses to Questions

**Comments to the Author**

1. If the authors have adequately addressed your comments raised in a previous round of review and you feel that this manuscript is now acceptable for publication, you may indicate that here to bypass the “Comments to the Author” section, enter your conflict of interest statement in the “Confidential to Editor” section, and submit your "Accept" recommendation.

Reviewer #1: All comments have been addressed

Reviewer #2: All comments have been addressed

Reviewer #3: All comments have been addressed

2. Is the manuscript technically sound, and do the data support the conclusions?

Reviewer #1: Partly

Reviewer #2: Yes

Reviewer #3: Yes

3. Has the statistical analysis been performed appropriately and rigorously? 

Reviewer #1: I Don't Know

Reviewer #2: Yes

Reviewer #3: Yes

4. Have the authors made all data underlying the findings in their manuscript fully available?

Reviewer #1: Yes

Reviewer #2: Yes

Reviewer #3: Yes

5. Is the manuscript presented in an intelligible fashion and written in standard English?

Reviewer #1: Yes

Reviewer #2: Yes

Reviewer #3: Yes

6. Review Comments to the Author

Reviewer #1: This research is interesting and provide some significant results. But the sample size is relatively small. And in the revised manuscript Ref. 12 should be “J Neurol Sci. 1994 Jul;124 Suppl:96-107”.

Reviewer #2: The authors have replied to my observations and according to me their answers and changes are satisfactory.

Reviewer #3: The authors have addressed all of my comments and the comments raised by the other reviewers; I have nothing more to add

7. PLOS authors have the option to publish the peer review history of their article (what does this mean?). If published, this will include your full peer review and any attached files.

Reviewer #1: No

Reviewer #2: No

Reviewer #3: No

---

## [Editor Report · Acceptance letter]

12 Nov 2021

PONE-D-21-22395R1 

Amyotrophic lateral sclerosis: Correlations between fluid biomarkers of NfL, TDP-43, and tau, and clinical characteristics 

Dear Dr. Kasai:

I'm pleased to inform you that your manuscript has been deemed suitable for publication in PLOS ONE. Congratulations! Your manuscript is now with our production department. 

Kind regards, 

on behalf of

Dr. Weidong Le 

Academic Editor

PLOS ONE